# Differential Induction Pattern Towards Classically Activated Macrophages in Response to an Immunomodulatory Extract from *Pleurotus ostreatus* Mycelium

**DOI:** 10.3390/jof7030206

**Published:** 2021-03-11

**Authors:** Gabriel Llauradó Maury, Humberto J. Morris-Quevedo, Annick Heykers, Ellen Lanckacker, Davie Cappoen, Peter Delputte, Wim Vanden Berghe, Zelene Salgueiro, Paul Cos

**Affiliations:** 1Centre of Studies for Industrial Biotechnology (CEBI), University of Oriente, Ave. Patricio Lumumba s/n, Reparto Jiménez, Santiago de Cuba 90500, Cuba; jquevedo@uo.edu.cu; 2Faculty of Applied Sciences, University of Camagüey, Camagüey 74650, Cuba; 3Laboratory for Microbiology, Parasitology and Hygiene (LMPH), Department of Pharmaceutical Science, Faculty of Pharmaceutical, Biomedical and Veterinary Sciences, University of Antwerp, Universiteitsplein 1, 2610 Antwerp, Belgium; annick.heykers@uantwerpen.be (A.H.); ellen.lanckacker@yahoo.com (E.L.); davie.cappoen@uantwerpen.be (D.C.); peter.delputte@uantwerpen.be (P.D.); 4Laboratory of Protein Chemistry, Proteomics and Epigenetics Signalling (PPES), Department of Biomedical Sciences, Faculty of Pharmaceutical, Biomedical and Veterinary Sciences, University of Antwerp, Universiteitsplein 1, 2610 Antwerp, Belgium; wim.vandenberghe@uantwerpen.be; 5Pharmaceutics Laboratories, BioCubaFarma, Ave 5ta, Distrito 30 de Noviembre, Santiago de Cuba 90400, Cuba; zelenesalgueiro83@gmail.com

**Keywords:** immune-activating, macrophage, mushroom, mycelium, *Pleurotus ostreatus*

## Abstract

*Pleurotus ostreatus* mushroom preparations have been investigated because of their ability to modulate the immune function. However, there is still no consensus regarding the activation and polarizing effect on macrophages by *Pleurotus*-derived bioproducts. This study examined the immune-activating effect of a mycelium-derived *P. ostreatus* aqueous extract (HW-Pm) on macrophage functions, by means of the determination of nitric oxide (NO) production, the mRNA expression of inducible nitric oxide synthase (iNOS), Arginase-1 and FIZZ and the cytokine levels. The phagocytic activity and the activation of NF-κB in U937 reporter cells were also investigated. No cytotoxicity was observed in macrophages treated with HW-Pm (IC_50_ > 1024 μg/mL) by the resazurin test. HW-Pm induced high levels of NO production and iNOS expression in macrophages. In contrast, HW-Pm did not induce Arginase-1 and FIZZ mRNA expressions. The mushroom extract increased TNF-α and IL-6 production and the phagocytic function in murine macrophages. It also stimulated the activation of the NF-κB promoter. The *P. ostreatus* mycelium extract has a potential application as a natural immune-enhancing agent, by targeting macrophage activation towards the classically activated subset and stimulating macrophage-mediated innate immune responses.

## 1. Introduction

Functional foods and bioactive components from natural sources are emerging as complementary treatments for immunological disorders [1,2,3]. Edible and medicinal mushrooms are considered to be a new wave of functional bio-ingredients or biologically active substances with high nutritional and functional value [4,5].

Both mycelium and mushroom fruiting bodies constitute an untapped source of compounds with a wide spectrum of biological activities [6]. Studies with mushroom-derived extracts or compounds have displayed the potential to activate several host immune mechanisms, via stimulation of both innate and adaptive immune pathways [7]. Bioactive polysaccharides, generally known as β-glucans, isolated from *Lentinus edodes*, *Agaricus bisporus*, *Ganoderma lucidum*, *Schizophyllum commune* and *Pleurotus* sp. mushrooms have been reported to trigger biological effects by binding to cell surface receptors in macrophage-like leukocytes [8,9,10]. The main receptor for β-glucan is dectin-1, and the immunomodulatory activity includes the stimulation of cytokine and chemical mediators’ production and the activation of phagocytic and tumoricidal actions [8,11].

The monocyte–macrophage system involves phagocytic cells that play an important role in the immune system by participating as an essential link between innate and acquired immune responses [12]. They exhibit a wide range of biological actions, including antigen presentation, elimination of pathogens, anti-tumoral properties and a key role in wound healing and inflammation [13,14].

Classically activated macrophages can be triggered by several factors. The diversity of the stimuli includes bacterial and fungal components, cytokines, chemical mediators [14,15] and some active substances from plants and mushrooms [16,17]. After being stimulated, the phagocytic cells produce and release cytotoxic mediators and pro-inflammatory cytokines, such as nitric oxide (NO) and TNF-α, IL-1β and IL-6, respectively [18,19].

It has been well-documented in both preclinical and clinical studies that immunodeficiencies acquired by malnutrition and neoplasia impair the number of monocytes and macrophages and their phagocytic action [20,21]. Antigen presentation, as well as the production of reactive oxygen and nitrogen intermediates by macrophages, is reduced [22]. For this reason, the restoration of these mechanisms to normal levels could be useful to combat the immunodepressive status.

The *Pleurotus* genus belongs to Asian folk medicine and includes a large group of edible mushrooms with pharmacological properties [9,23,24]. Traditional mycelium and fruiting bodies-derived preparations from *Pleurotus* species, like decoctions or hot water extracts, have commonly been used as complementary remedies to treat several human diseases and pathological conditions, such as wounds, cancer, cytostatic treatment-related secondary effects and immunological disorders [25,26,27].

In a previous study, we reported the in vitro immunomodulating activity of a hot-water extract from mycelium of the oyster mushroom, *Pleurotus ostreatus*. The mycelia extract enhanced the acid phosphatase lysosomal activity in murine peritoneal macrophages. We hypothesized that macrophage activation might be related to binding of some mushroom component(s) to receptors found on phagocytic cells [28].

In the literature, there is no consensus on the cytokine profile and mediators induced in leukocytes by *Pleurotus* mushroom preparations and their active substances. Both in vitro pro-inflammatory and anti-inflammatory effects have been found for some *Pleurotus*-derived bioproducts. This raises questions about the relations between mycochemical composition, mushroom growth stage and modulation of macrophage function [19,29,30,31].

The aim of the present study was to investigate the in vitro immunomodulating effects of *P. ostreatus* mycelium extract (HW-Pm) by measuring nitric oxide production, mRNA expression of iNOS, the release of TNF-α and IL-6 cytokines and the phagocytic function in RAW 264.7 murine cells. The results on the immunomodulatory effects will contribute to promoting mycelial-derived products in the mushroom nutraceutical and pharmaceutical market, in which fruiting bodies-derived products are the focus of much of this interest.

## 2. Materials and Methods

### 2.1. Materials and Reagents

Lipopolysaccharide (LPS) from *Escherichia coli* (0128:B12), PMA (Phorbol 12-Myristate 13-Acetate), endotoxin-free ultra-pure water, FeO_4_S·7H_2_O, tamoxifen and resazurin sodium salt (7-Hydroxy-3H-phenoxazin-3-one-10-oxide) were purchased from Sigma-Aldrich (St. Louis, MO, USA). Dulbecco’s modified Eagle medium (DMEM), Dulbecco’s phosphate-buffered saline (DPBS), RPMI-1640 medium and fetal calf serum (FCS) were from Gibco^®^ (New York, NY, USA). The Griess reagent kit was purchased from Molecular Probes^®^ (Eugene, OR, USA) and hygromycin from Thermo Fisher. Electron paramagnetic resonance (EPR)-grade water and Krebs HEPES were purchased from NOXYGEN Science Transfer and Diagnostics GmbH (Elzach, Germany). DETC (diethyldithiocarbamic acid sodium salt trihydrate) and L-NAME (N5-(imino(nitroamino)methyl)-L-ornithine, methyl ester, monohydrochloride) were purchased from ALEXIS Biochemicals (New York, NY, USA) and Cayman Chemical (Ann Arbor, MI, USA).

### 2.2. Pleurotus ostreatus Strain and Extract Preparation

A commercially cultivated strain of *Pleurotus ostreatus* (Jacq:Fr.) Kumm (Pleurotaceae) (code number: CCEBI-3024) deposited at the Culture Collection of the Centre of Studies for Industrial Biotechnology (CEBI, Cuba) was used. The strain was maintained on slants with a solid medium of potato dextrose agar (PDA) incubated at 5 °C. The Cuban Research Institute of Sugar Cane By-Products (ICIDCA, Havana City, Cuba) kindly provided this mushroom strain, supplied by A. Ginterová (Strain Collection of VUKPS, Bratislava, Slovakia).

A standardized procedure to obtain the mushroom extract was carried out, following the methodology described in a patent protocol deposited at the Cuban Office of Industrial Property [32], and used to develop mushroom preparations with in vitro anti-proliferative effect on NB4 human leukemia cells [33] and in vitro antimicrobial and complement/macrophage-stimulating activities [28].

The hot water extract from *P. ostreatus* (HW-Pm) was prepared by submerged fermentation of mushroom mycelium. Briefly, mycelium was inoculated in Erlenmeyer flasks, which contained YPG medium (yeast–peptone­–glucose). The flasks were incubated at 27 °C with continuous stirring at 100 rpm (MIZARD 2001 Shaker, RETOMED, Santiago de Cuba, Cuba) for 15 days. Afterward, mushroom mycelium was filtered through Whatman no. 4 paper and washed exhaustively with distilled water to discard any residue of culture medium, which would contain soluble polysaccharides secreted by fungal cell (not tested here). Cell biomass—200 g wet weight/L of distilled water, ratio of 5:1 water (mL)/ biomass (g)—was then extracted at 95 °C for 10 h, and the resulting crude extract was collected by centrifugation at 4000 rpm. Then, the extract was dried, suspended in endotoxin-free ultra-pure water, filtered with 0.2 µm bacteriological filters, aliquoted and kept frozen at −20 °C until use.

The yield of dried HW-Pm was 10% as a percentage weight of the starting fresh mushroom. The main biochemical composition identified in HW-Pm was 70.8% carbohydrates [34] and 16.6% proteins [35]. Similar ranges of carbohydrates (70.5–76.8%) and proteins (12–15%) have been determined in previous studies [28,33]. Nucleic acids (0.11%), minerals (5%), β-1,3-1,6-glucans (1.5%) and total phenolic compounds (38 ± 5 mg/g extract) have also been reported in similar batches of mycelia extracts [28,33]. These values are in agreement with the quality assurance criteria for HW-Pm, in which coefficient variations between batches are less than 3% (CEBI-UO Technical Procedure for Obtaining HW-Pm, 2011).

The endotoxin levels were examined in HW-Pm by the limulus amoebocyte lysate test (LAL assay according to CAPE COD, Inc., East Falmouth, MA, USA). The HW-Pm contained 0.019 EU/mL of endotoxin. A sample of demineralized water used to prepare the mushroom extract was also tested for endotoxins (0.017 EU/mL).

### 2.3. Cell Culture

RAW 264.7 macrophages were purchased from ATCC (American Type Tissue Culture Collection, Manassas, VA, USA) and maintained at 37 °C, 5% CO_2_ atmosphere in DMEM medium supplemented with 10% FCS, 2% l-glutamine and 4.5 g/L d-glucose.

U937-3xB-LUC cells, which were kindly provided by professor Wim Vanden Berghe, were maintained in RPMI-1640 supplemented with L-glutamine (2 mM), hygromycin (75 mg/mL) and 10% FCS at 37 °C and 5% CO_2_.

#### Cell Viability Assay

The effect of HW-Pm on RAW 264.7 cell viability was determined by the resazurin dye reduction test (Appendix A). Briefly, 200 µL of cell suspension (5 × 10^5^ cells/mL) was added to a 96-well plate and incubated at 37 °C in a humidified atmosphere with 5% CO_2_. After 24 h, the RAW cells were washed twice with 200 µL of DPBS, and HW-Pm extract (2, 4, 8, 16, 32, 64, 128, 256, 512 and 1024 µg/mL) in DMEM (FCS-free) was added into each well and incubated at 37 °C, 5% CO_2_. After 24 h, 50 µL of resazurin solution (2.2 µg/mL) was added, and the fluorescence was measured after 4 h (λ excitation 550 nm, λ emission 590 nm; TECAN GENios microplate reader; TECAN Group Ltd., Männedorf, Switzerland). A medium without extract was included as a negative control. Two independent experiments were performed for the experiments and the samples were tested by triplicate in each one. Tamoxifen was included as a reference control drug for cytotoxicity.

### 2.4. Nitrites and Cytokine Determination

The extracellular nitrite accumulation was measured by the Griess reaction (Appendix A). Macrophages (5 × 10^5^ cells/mL in a 24-well plate) were treated with HW-Pm extract (100 and 500 µg/mL) during 24 h. Afterwards, 150 µL of culture supernatant was collected and transferred to 96-well plate filled with 130 µL of demineralized water. After adding 20 µL of Griess reagent, samples were incubated for 30 min protected from light, and the absorbance was measured at 540 nm. A standard calibration curve was set up by diluting the nitrite standard solution of the kit.

The cytokine levels in supernatants were determined by mouse TNF-α and IL-6 immunoassay Quantikine^®^ ELISA Kits (R and D Systems Inc., Minneapolis, MN, USA). LPS (100 ng/mL) and IFN-γ (5 ng/mL) were used as positive controls for stimulation.

### 2.5. Intracellular NO Determination by EPR Spectroscopy

RAW cells (5 × 10^5^ cells/mL) were stimulated in a T25 culture flask for 24 h with LPS (100 ng/mL) and IFN-γ (5 ng/mL) as positive controls as well as with HW-Pm extract (100 and 500 µg/mL). The cells were incubated with the spin trap [Fe(DETC)_2_] (iron diethyldithiocarbamate) for 1 h [36]. After removal of the supernatant, resuspended cells were brought into a 1 mL syringe and frozen in liquid nitrogen. All results were given as the NO spin-trap adduct concentration that was reached after 1 h accumulation of the [NO-Fe(DETC)_2_] complex in stimulated cells minus the concentration in non-stimulated (control) cells (Appendix A).

EPR measurements were performed in the bench-top EPR Spectrometer MiniScope MS200 with the temperature controller TC-H02 (Magnettech, Germany). The signal intensity was expressed in terms of peak height. The peak height ∆-Y is the arbitrary distance between the lowest and highest point of the curve on the Y-axis. The EPR spectrum-data processing was performed by Windows EPR spectrometer software v2.5.1 (Magnettech, Freiberg, Germany, 2003).

### 2.6. RNA Extraction and Quantitative Real-Time PCR

RAW 264.7 cells (3 mL, 1 × 10^6^ cells/mL) were seeded in 6-well plates and incubated with HW-Pm extract (500 and 100 μg/mL), LPS (100 ng/mL) and IFN-γ (5 ng/mL) for 24 h at 37 °C and 5% CO_2_. The supernatant was discarded, and the cells were collected in fresh DMEM and centrifuged to 2000 rpm for 10 min at 4 °C. RNA-later (350 μL) was added to the pellet, and the suspension was maintained to −80 °C until use. The cells were lysed, and total RNA extraction was performed by using a mini-kit RNeasy^®^ Plus (QIAGEN^®^, Germantown, MD, USA). The purified RNA was quantified by spectrophotometry (NanoDrop, Wilmington, DE, USA) at 260 nm, and the purity of the extracted RNA was measured by the 260/280 index. The total RNA extracted was kept at −80 °C until use. In addition, a pool with all RNA samples was prepared (LPS-IFN-γ, IL-4 and control).

cDNA was synthesized from 0.7 µg aliquots of RNA using a High-Capacity cDNA RT kit (Applied Biosystems, Foster City, CA, USA) in a thermocycler (StepOnePlus™ Instrument, Foster City, CA, USA). Amplification was carried out with the following cycle profile: 1 cycle at 95 °C for 10 min (polymerase activation) and 45 cycles of 20 s at 95 °C (denaturation) and 1 min at 60 °C (annealing and extension) according to the manufacturer’s protocol. cDNA standards of HGPRT (hypoxanthine-guanine phosphoribosyltransferase), iNOS, Arg-1 and FIZZ were used to perform the calibration curve. Gene-expression levels were assessed and normalized using HGPRT as the internal control. The results were expressed as the ratio of quantity between the HGPRT gene and the iNOS, Arg-1 and FIZZ genes. Primers (HGPRT: assay ID Mm03024075_m1, iNOS: assay ID Mm00440502_m1, Arg-1: assay ID Mm00475988_m1 and FIZZ: assay ID Mm00445109_m1) were purchased from Applied Biosystems (Foster City, CA, USA) Reference Number: 4331182 (250 rxn)(Appendix A).

### 2.7. Phagocytosis Assay

Macrophages (5 × 10^5^ cells/mL) were incubated in 24-well plates with coverslips (24 h, 37 °C, 5% CO_2_). HW-Pm (at 100 and 500 μg/mL), LPS (100 ng/mL) and IFN-γ (5 ng/mL) were added to the wells. DMEM was used as a negative control. Carboxylate-modified polystyrene yellow–green fluorescent beads (Merck, Sigma-Aldrich, Darmstadt, Germany), diluted to 1/500 in DMEM, were added (300 μL per well) to the RAW 264.7 cells and kept at 37 °C for 6 h. Macrophages maintained at 4 °C during the assay were used as a negative control of phagocytosis. Finally, macrophages were washed to remove un-phagocytozed beads, fixed with 4% paraformaldehyde (Merck) and permeabilized using 0.5% Triton X-100 (Sigma-Aldrich). Macrophages were further stained with Texas Red^®^-X phalloidin (Invitrogen^TM^, Waltham, MA, USA) to stain F-actin, and cell nuclei were visualized with 4′,6-diamidino-2-phenylindole (DAPI) (Sigma-Aldrich) under a fluorescence microscope (ZEISS, Carl Zeiss AB, Göttingen, Germany). The experiment was performed in triplicate, and phagocytozed particles of 50 different macrophages were counted (Appendix A).

### 2.8. NF-κB Reporter Gene In Vitro Assay

The assay was performed using a human monocytic cell line, stably transfected with a luciferase reporter containing three NF-κB-binding sites (U937-3xB-LUC cells) [37] to evaluate the modulatory activity of HW-Pm on NF-κB activation. The U937-3xB-LUC cells (5 × 10^5^ cells/mL) were maintained following the conditions aforementioned. Afterwards, the monocytes were transferred to a new medium without antibiotics in 96-well plates and incubated for 24 h with PMA (10 ng/mL) to differentiate the cells into macrophages. The medium with PMA was replaced, and the cells were stimulated with HW-Pm extract (100–500 μg/mL) for 24 h. Cells incubated with LPS (100 ng/mL) were used as a positive control of stimulation. The luciferase reporter-gene expression was determined by cell lysis and luciferase (luc) in vitro assays, developed according to the protocol Steady-Glo^®^ Luciferase Assay System of Promega (Promega, Madison, WI, USA). Briefly, equal volumes of the reagent and culture medium were mixed and maintained at room temperature prior the experiment. Afterward, the mixture was added to each well 5 min before the luminescence measurement to allow sufficient cell lysis. L-NAME (100 µM) was used as a reference control to suppress the LPS-derived pro-inflammatory stimulus (S6).

### 2.9. Statistical Analysis

Statistical analysis was performed using the statistical software package, GraphPad Prism 7 (Windows, v7.04, 2017). All results were statistically analyzed and expressed as the arithmetic means ± standard deviation (SD). A one-way ANOVA test, followed by the Tukey test, was applied to determine the significance of differences between groups. Differences at *p* ≤ 0.05 were considered as significant.

## 3. Results

### 3.1. Effect of HW-Pm on RAW 264.7 Macrophages Cell Viability

The RAW 264.7 macrophages were treated with several doses of HW-Pm ranging from 2 to 1024 µg/mL for 24 h. HW-Pm was not cytotoxic to RAW 264.7 cells (Figure 1). Therefore, we used two non-toxic concentrations, i.e., 100 and 500 µg/mL, in further experiments.

### 3.2. Effect of HW-Pm on Nitric Oxide Production in RAW 264.7 Macrophages

RAW 264.7 cells treated with HW-Pm at 100 and 500 µg/mL resulted in an increased nitrite production, which was statistically comparable with the LPS-stimulated positive control. Unstimulated macrophages exposed to the medium alone (negative control) produced undetectable levels of nitrite (*p* < 0.05) (Figure 2A).

EPR is a robust and sensitive method to detect NO radicals as stable [NO-Fe(DETC)_2_] radicals. RAW 264.7 macrophages stimulated with HW-Pm at 100 and 500 µg/mL showed a higher EPR signal compared to unstimulated macrophages. The values, expressed as arbitrary units (AU), were similar to the cells stimulated with lipopolysaccharides and IFN-γ (*p* < 0.05) (Figure 2B).

### 3.3. HW-Pm Enhanced iNOS mRNA Expression in RAW 264.7 Macrophages

To elucidate if the NO production by the *Pleurotus* extract was due to the upregulation of iNOS mRNA expression, real-time PCR assays were performed. The iNOS mRNA-expression levels were significantly elevated in macrophages exposed to HW-Pm and, similarly, to cells treated with LPS and IFN-γ (*p* < 0.05) (Table 1); particularly, HW-Pm at 500 µg/mL exhibited a strong activity. These results suggest that the stimulatory effect of the extracts on NO production was mediated by the induction of the iNOS gene. Expression of alternatively activated macrophages or M2 subset, Arg-1 and FIZZ anti-inflammatory genes [38], were not detected.

### 3.4. Effect of HW-Pm on TNF-α and IL-6 Production in RAW 264.7 Macrophages

The stimulatory effect of HW-Pm on TNF-α and IL-6 production in macrophages was measured by an ELISA assay. Macrophages incubated with HW-Pm for 24 h produced significant levels of TNF-α and IL-6 cytokines (*p* < 0.05), compared to the classical stimulation with LPS and IFN-γ (Figure 3). The HW-Pm-treated cells at 500 µg/mL showed the highest stimulatory effect in this cell line, indicating a dose-dependent response for cytokine production.

### 3.5. Phagocytic Activity of RAW 264.7 Macrophages Is Induced by HW-Pm

The HW-Pm extract significantly increased the in vitro phagocytic ability of murine macrophages (Table 2) (*p* < 0.05). The extract at a dose of 500 µg/mL showed the highest stimulation of macrophage phagocytosis, even compared with the positive control (LPS and IFN-γ stimulation) (Figure 4).

### 3.6. NF-κB Activation

Stimulation with HW-Pm for 24 h slightly enhanced the activity of NF-κB in U937-3xB-LUC cells. The study evidenced that the extract at 500 µg/mL showed statistically higher values of relative NF-κB activity (*p* < 0.05) (Figure 5) in comparison to unstimulated control cells. Cells stimulated with LPS and IFN-γ were used as a positive control for the activation of the NF- κB pathway. L-NAME at 100 µM significantly decreased NF-κB activation in LPS- and IFN-γ-stimulated cells.

## 4. Discussion

Several functional preparations rich in active substances are being formulated to treat various diseases in combination with conventional medications. Both fruiting bodies and mycelia of edible mushrooms are largely known to possess active substances that exert medicinal effects on the host’s immune system [7]. Therefore, the modulation of the immune system by mushroom extracts or isolated components may emerge as an important strategy to combat some immunopathologies.

Numerous reports suggest that mycochemicals from mushrooms can act by elevating or regulating the host immune response, both in humans and animal models. The process includes the activation and/or regulation of different mechanisms in dendritic cells, natural killer (NK) cells, T cells and mainly macrophages [7,8,39].

In the case of the *Pleurotus* genus, various studies have highlighted its potential to modulate immune action. However, different biopreparations and semi-purified fractions of these mushrooms have exhibited both in vitro pro-inflammatory and anti-inflammatory effects in T lymphocytes and macrophages [16,19,31,39]. For that reason, the present work was undertaken to elucidate the immune-enhancing potential of a *Pleurotus* mycelia-derived extract on RAW 264.7 murine macrophages. Here, we showed that (a) HW-Pm stimulated NO production in murine macrophages without affecting the cell viability; (b) the release of NO was upregulated by the expression of iNOS in macrophages; (c) the production of the pro-inflammatory cytokines, TNF-α and IL-6, was induced in macrophages after stimulation with HW-Pm; and (d) the *Pleurotus* extract also increased the phagocytic capacity of macrophages and the activation of NF-κB in U937-3xB-LUC cells.

Because macrophages have a significant role in immunological surveillance, many experiments have aimed to modulate their functions through activation by immunoenhancing substances. Previously, we showed that an aqueous fraction from *Pleurotus* mycelia increased the macrophage function by a dose-dependent augment of lysosomal enzymes. It was suggested that the activation was related to some components, i.e., β-d-glucans, present in HW-Pm that can bind to receptors on the macrophage cell surface [28].

One of the most consistent markers to measure classical activation in murine macrophages is NO production [12,18]. Since M1 macrophages are implicated in antimicrobial and cell-mediated immune responses, NO appears to be essential in these functions [40,41]. Therefore, *Pleurotus* extract could improve innate immunity through immunostimulating mycochemicals by increasing host resistance to microbes. The polarizing effect exerted by HW-Pm towards M1 macrophages involved iNOS participation, presumably driven by NF-κB activation. In this sense, we found that incubation of the cell line U937-3xB-LUC, differentiated to macrophages, with HW-Pm resulted in NF-κB activation. The human monocytic cell line was stably transfected with a luciferase reporter gene that contains three NF-κB-binding sites. This cell line has been used as a high throughput cell-based screening system to evaluate the activation or inhibition of NF-κB [42]. Here, we hypothesized that the production of pro-inflammatory mediators and iNOS gene expression in macrophages were probably mediated by the NF-κB family.

The polarizing effect of macrophages by *Pleurotus* fractions has also been reported in another study [30]. A β-glucan (PCPS) obtained from an extract of *Pleurotus cornucopiae* fruiting bodies showed a stimulatory effect on both THP-1 human and peritoneal murine macrophages. The PCPS fraction increased mRNA expression of TNF-α and IL-1β pro-inflammatory cytokines in macrophages. However, the same study evidenced that the *P. cornucopiae* extract (containing mainly PCPS) did not augment the expression levels of IFN-γ mRNA in murine splenocytes, which would be attributed to other immune regulative components in the mushroom.

Until now, the in vitro evidence regarding macrophage polarization by *Pleurotus* mushrooms has not been consistent. Jedinak et al. [19] reported that an oyster mushroom concentrate reduced the LPS-induced release of pro-inflammatory cytokines, TNF-α and IL-6, in RAW 264.7 macrophages. The anti-inflammatory effect of the concentrate was attributed to both the presence of different amino acids, vitamins and water-soluble glucans. An ethanol extract prepared from basidiocarps of *Pleurotus giganteus* (Berk. Karun and Hyde) also showed significant suppression of inflammatory markers in a murine macrophage cell line [30]. The authors described a strong reduction of LPS- and H_2_O_2_-induced pro-inflammatory mediators’ production such as iNOS, STAT and cyclooxygenase-2 (COX-2) despite having a highly activated NO production. They concluded that the inhibition of iNOS expression in RAW 264.7 cells was significant, although the release of NO was only partially reduced. Finally, the anti-inflammatory effect of *Pleurotus* giganteus was not correlated with the mushroom chemical composition in the study.

In contrast, we found that *P. ostreatus* enhanced the production of pro-inflammatory cytokines by macrophages. It is generally accepted that classically activated macrophages show an upregulation of cell adhesion molecules, such as intercellular adhesion molecule-1 (ICAM-1), and also release a number of different inflammatory mediators, including NO, TNF-α and IL-6 proteins [12]. As described previously, the release of cytokines involved in inflammation could also be associated with the presence of some innate immunity receptors that recognize β-glucans. In a previous paper, we found β-1,3-1,6-glucans at 1.5% of the total dry mass in a hot water extract from *Pleurotus* mycelia [33]. We also demonstrated the possible in vitro activation of the complement classical pathway by two polysaccharide-enriched fractions isolated from *Pleurotus* mycelia extract. We hypothesized that the mechanism of action was based on the binding ability of antibodies to β-glucans, thereby forming immune complex-like substances [28].

In addition, a clinical trial corroborated the potential effect of oyster mushroom to polarize activation towards a Th1 or classical activation. *Pleurotus* extract probably augmented the immune response through Th1-phenotype potentiation as the macrophage-IL-12–IFN-γ pathway [43].

An overview of both anti-inflammatory and/or pro-inflammatory effects of oyster mushrooms links the mycochemical profiling with its biological action. Methanol and ethanol extracts, containing high levels of antioxidants, have shown an enormous potential to inhibit LPS-induced inflammation. Polyphenol-enriched extracts from *Pleurotus florida* and *Pleurotus pulmonarius* fruiting bodies reduced the nitrite accumulation in macrophages. Conversely, the aqueous fractions showed a lower anti-inflammatory effect [44].

Our study revealed that HW-Pm did not stimulate Arg-1 and FIZZ mRNA expression. The mushroom extract did not polarize macrophages towards the alternatively activated subset. FIZZ and Arg-1 are considered robust and reliable markers of the anti-inflammatory or M2 macrophage subset [38,45].

Twenty-four hours of treatment with HW-Pm stimulated the phagocytic activity of RAW 264.7 cells. This finding also confirmed the results previously described that some myco-compounds present in HW-Pm could bind to cell receptors to induce the release of cytokines and chemical mediators [14]. It has been observed that after the PAMPs (like fungal polysaccharides) recognition by phagocytic receptors, a variety of cell signalling cascades can be activated [14].

Fungal extracts can modulate the local immune system at the intestinal tract, exerting systemic immune activation. A hypothesis defends the passages of mushroom polysaccharides through the gap junctions in the intestinal epithelial membrane, internalization through intestinal M and macrophage cells and absorption after binding with toll-like receptors on the intestinal lumen [7,46]. We previously revealed the activation of Kupffer cells and splenic macrophages after the oral administration of a crude extract of *Pleurotus ostreatus* fruiting bodies in malnourished mice, which might be related to partial absorption of mycochemicals such as polysaccharides [47]. In addition, mushrooms are considered excellent prebiotics because of their ability to exert beneficial effects on gut microbiota, conferring health maintenance to the host [48]. However, the mechanism by which other mycochemicals are absorbed and stimulate the immune system after enteral administration is still unclear.

In sum, the obtained results provide new insights to elucidate the modulating profile towards anti- or pro-inflammatory responses of *Pleurotus* preparations (pro-inflammatory, in this case). So far, potential explanations to clarify the differences in this subject may be linked to the mycochemical composition of bioproducts, the correlation of in vitro and in vivo doses, the type of resident macrophages or cell lines selected and the type and/or duration of stimuli. Fermentation techniques could also influence the diversity of myco-compounds in mushroom extracts. In this context, liquid-submerged fermentation has been proposed to be more uniform and reproducible for the development of medicinal substances compared to solid-state fermentation [27,49]. As it was aforementioned, we previously informed the presence of β-1,3-1,6-glucans and secondary metabolites in a similar extraction protocol from *Pleurotus* mycelia obtained under liquid-submerged fermentation [33]. Nevertheless, a detailed study of a single treatment of *Pleurotus* mycochemical components will be necessary to completely comprehend how these components can modulate macrophage activation.

## 5. Conclusions

This study demonstrated the immunomodulatory potential of a mycelia-derived *P. ostreatus* extract in RAW 264.7 murine macrophages. HW-Pm led to the activation of the pro-inflammatory profile of macrophages. The NO production, the release of TNF-α and IL-6 cytokines and the activation of phagocytic mechanisms suggest that HW-Pm treatment promotes polarization towards classically activated macrophages. Reporter-gene studies reveal a possible role for NF-κB activation as a mechanism-of-action. Altogether, these results support the use of functional formulations from *P. ostreatus* as a complementary therapy to manage immune dysfunctions and to maintain immune homeostasis.

## Figures and Tables

**Figure 1 jof-07-00206-f001:**
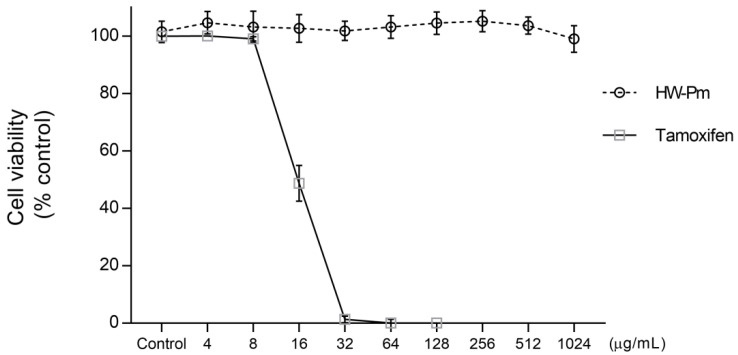
Effect of HW-Pm on RAW 264.7 murine macrophage cells’ viability. The effect of HW-Pm on the viability of RAW 264.7 cells was determined by the resazurin assay. All values are expressed as the arithmetic mean ± SD of six replicates. HW-Pm is the hot water extract from *P. ostreatus* and tamoxifen is a reference control drug for cytotoxicity.

**Figure 2 jof-07-00206-f002:**
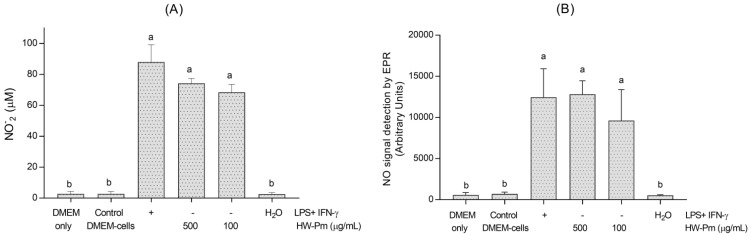
Effect of HW-Pm on nitric oxide production by RAW 264.7 macrophages. The nitric oxide production was measured by Griess reagent (**A**) and by EPR (**B**). LPS (100 ng/mL) and IFN-γ (5 ng/mL) were used as positive controls for stimulation. Means without the same letter are significantly different at the 5% level according to a one-way ANOVA test followed by a Tukey test, (*n* = 6) respectively.

**Figure 3 jof-07-00206-f003:**
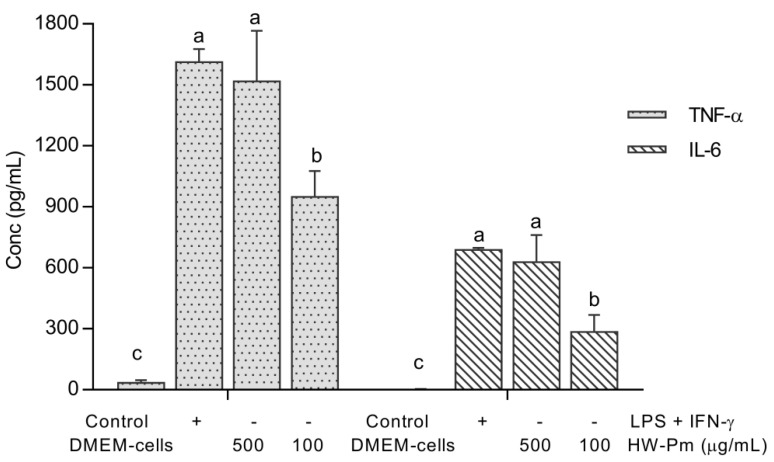
Effect of HW-Pm on TNF-α and IL-6 production in RAW 264.7 macrophages. The cytokine levels were determined by mouse TNF-α and IL-6 immunoassay Quantikine^®^ ELISA Kits. Means without the same letter are significantly different at the 5% level according to a one-way ANOVA test followed by a Tukey test (*n* = 4).

**Figure 4 jof-07-00206-f004:**
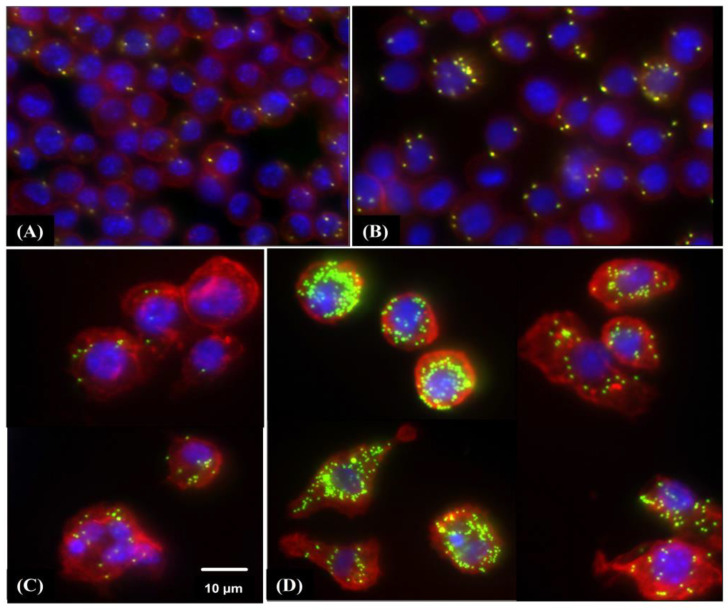
Immunofluorescence staining of RAW 264.7 macrophages stimulated by HW-Pm. Macrophages were incubated with HW-Pm (100 and 500 µg/mL) and LPS (100 ng/mL) and IFN-γ (5 ng/mL), respectively. Carboxylate-modified polystyrene yellow–green fluorescent beads were added to the cells. Macrophages were fixed with 4% paraformaldehyde, permeabilized using 0.5% Triton X-100 and further stained with Texas Red^®^-X phalloidin (Invitrogen^TM^, Waltham, MA, USA) to stain F-actin, and cell nuclei were visualized with 4′,6-diamidino-2-phenylindole (DAPI) under a fluorescence microscope. Representative images of each group (Scale bar 10 µm): (**A**) Control (DMEM), (**B**) LPS–IFN-γ, (**C**) HW-Pm 100 µg/mL and (**D**) HW-Pm 500 µg/mL.

**Figure 5 jof-07-00206-f005:**
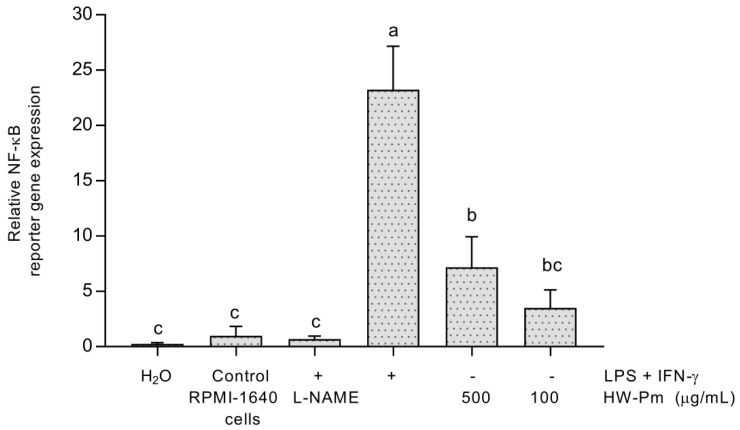
Effect of HW-Pm on NF-κB activation. The assay was performed using a human monocytic cell line, stably transfected (U937-3xB-LUC reporter cell) with a luciferase reporter containing three NF-κB-binding sites. LPS (100 ng/mL) and IFN-γ (5 ng/mL) were used as positive controls for stimulation. L-NAME was tested at 100 µM. Means without the same letter are significantly different at the 5% level according to a one-way ANOVA test followed by a Tukey test (*n* = 3).

**Table 1 jof-07-00206-t001:** Effect of HW-Pm on iNOS, Arg-1 and FIZZ mRNA expression in RAW 264.7 macrophages.

	M1 Subset	M2 Subset	
	iNOS	Arg-1	FIZZ
Control	0.44 ± 0.04 ^c^	0.01	- (nd)
LPS + IFN-γ	5.04 ± 0.59 ^ab^	0.03	-
HW-Pm (500 µg/mL)	5.98 ± 0.51 ^a^	0.01	-
HW-Pm (100 µg/mL)	4.17 ± 0.54 ^b^	0.03	-

The results are expressed as the ratios of quantity (ng/mL) between HGPRT gene and the iNOS, Arg-1 and FIZZ genes. LPS (100 ng/mL) and IFN-γ (5 ng/mL) were used as positive controls for stimulation. Means ± SD without the same letter are significantly different at the 5% level according to a one-way ANOVA test followed by a Tukey test (*n* = 3). (nd): not detected.

**Table 2 jof-07-00206-t002:** Effect of HW-Pm on phagocytosis of fluorescents beads by RAW 264.7 macrophages.

	Number of Beads/50 Cells	Phagocytic Ratio (%)
Control	72.3 ± 4.8 ^d^	42 ± 4.3 ^c^
LPS + IFN-γ	357 ± 6.8 ^b^	58.3 ± 3.9 ^b^
HW-Pm (500 µg/mL)	454 ± 30.8 ^a^	75.3 ± 2.5 ^a^
HW-Pm (100 µg/mL)	239 ± 13.9 ^c^	60 ± 4.9 ^b^

LPS (100 ng/mL) and IFN-γ (5 ng/mL) were used as positive controls for stimulation. Means ± SD without the same letter are significantly different at the 5% level according to a one-way ANOVA test followed by a Tukey test (*n* = 3).

## Data Availability

Not applicable.

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
