# Peer review of "Differential Induction Pattern Towards Classically Activated Macrophages in Response to an Immunomodulatory Extract from Pleurotus ostreatus Mycelium"

_jof, 2021, doi:10.3390/jof7030206_

Round 1

Reviewer 1 Report

This is an interesting, well-written and documented manuscript. The experimental work is sound investigating a critical aspect of edible fungi's biological actions, immunomodulation. 

Author Response

  • Reviewer 1

English language and style are fine/minor spell check required

R: The reviewer indicated that minor changes in the English language are required. Therefore, we thoroughly revised the manuscript for the English language, which can be viewed in the revised manuscript with track changes. We also verified the author names and a modification was done, which can be also viewed in the revised manuscript with track changes.

Exact change in author names: “Vanden Berghe, W”.

Reviewer 2 Report

Llaurado et al examine the effect of addition of a mushroom aqueous extract on macrophage activation.  The extract promoted inflammatory responses from macrophage cell lines that included enhanced phagocytosis, increased RNS production, and pro-inflammatory cytokine production.  The immunostimulatory effect of fungal extracts on macrophages is not new, but this study defines the specific effects of a P. ostreatus extract.

The main limitations of the study are (1) the use of a crude, uncharacterized extract and (2) no indications of variation due to extract preparation.  Variations in fungal growth (time, medium, etc.) can significantly influence the composition of the extract and therefore the effects on mammalian cells.  The authors only provide a high-level composition analysis of the extract without identifying the presence of specific molecules (such as beta-glucans which they hypothesize it contains).  A single extract preparation is used for all assays.  While an exhaustive survey of parameters is not necessary, the reproducibility of the extract and the main immunostimulatory effects should be included.

In addition, the authors do not provide sufficient details for the extracct preparation.  What was the growth medium for the submerged fermentation of P. ostreatus?  How long was the growth and at what temperature?  Was the mycelium washed before extraction with boiling water?  Was the extract combined with the culture supernatant?  How many biological replicates?

The discussion is overly long and is too repetitive of the results.  The authors should include in their discussion whether they think intestinal administration of the extract (i.e., feeding) would have systemic effects on an individual or whether the immunostimulatory effects would be confined to the intestinal tract (causing problems instead of systemic immune modulation).

Other corrections/suggestions

lines 26-30 remove methodological details from abstract

line 52  missing key reference showing Dectin-1 is the receptor for beta-glucans

line 75  give examples of diseases treated since the previous paragraph indicates overcoming immune deficiencies

line 115 give breif description of procedure.

line 127 missing reference for previous studies of carbohydrate/protein composition of extracts

line 143 define DPBS

line 143 why was serum omitted from the viability assay?  Serum should be included as it contains growth factors necessary for normal cell vitality

line 163 define "cells" as RAW cells (not to be confused with fungal cells)

line 165 define Fe(DETC)2

line 187 briefly define synthesis of cDNA, not just referencing a kit. What was the reverse transcription primer? how much RNA template? Define synthesis conditions (temperature, time, cycles)

line 198-199 what was the MOI for beads:macrophages?

line 138 indicate U937 cells, source, and general maintenance

line 219 briefly indicate main steps of luciferase assay (lysis, concentration of substrate, time for collection of luminescence).

line 260 provide reference for M1 vs M2 macrophages

line 301, 334  NF-kB itself is not increased in expression, just the NF_kB-activated reporter

lines 335-347 irrelevant discussion since the role of Dectin-1 or beta-glucans was never indicated in the experiments

lines 348-359 not discussion, just repetition of the results

Figure 2 - indicate panels A and B

Figure 2, 3, 5 specify what is "control" on the x-axis

Figure 4 define "control" treatment

Reviewer 3 Report

General Comments:

The authors have investigated the effects of mushroom-derived extract on innate immune responses in murine macrophage cell lines. Overall, current version of the manuscript is well written and organized. However, before publication, I would like to recommend the authors to correct several minor points as follows:

Minor points:

  1. Fig. 1. Although it was written in Materials and Methods section, please explain the abbreviation of HW-Pm and Tamoxifen in user-friendly manner. Thus, it will be much easier for future readers to read your paper after publication in this journal. For example, add “HW-PM is the hot water extract from P. ostreatus”, and Tamoxifen is “   “ and used as ---- control for Fig 1 legends.

Author Response

  • Reviewer 3

General Comments:

The authors have investigated the effects of mushroom-derived extract on innate immune responses in murine macrophage cell lines. Overall, current version of the manuscript is well written and organized. However, before publication, I would like to recommend the authors to correct several minor points as follows:

Minor points:

  1. Fig. 1. Although it was written in Materials and Methods section, please explain the abbreviation of HW-Pm and Tamoxifen in user-friendly manner. Thus, it will be much easier for future readers to read your paper after publication in this journal. For example, add “HW-PM is the hot water extract from P. ostreatus”, and Tamoxifen is “ “ and used as ---- control for Fig 1 legends.

We thank the editor for the opportunity to revise the manuscript according to the suggestions of the reviewers. We also want to thank the reviewers for the detailed revision. 

The reviewer indicated that Fig. 1 legend has to be improved. Therefore, we follow the suggestion made by the referee and we clearly state the abbreviations meaning and the reference control drug in the Fig.1 legend, which can be viewed below and in the revised manuscript with track changes.

  • Exact change/addition in Fig 1 legend: “HW-Pm is the hot water extract from ostreatus and Tamoxifen is a reference control drug for cytotoxicity”(line 272-273)
  • Exact addition: we also added in page number 4, first paragraph: “…reference control drug for…” (line 167)

Round 2

Reviewer 2 Report

The manuscript from Llaurado et al is significantly improved by attention to corrections and suggestions from reviewers.  The authors have detailed the extract preparation procedure adequately.

However, the main concern remains that the results presented represent a single extract preparation without any indication of variation.  The authors claim they use a standardized procedure which shows less than 3% variation (line 143), however what is measured for variation is not clear (i.e., simple protein concentration?  lipid profiles? etc.).  Extracts can vary in the composition of specific proteins or polysacchrides without showing variation in the total amount of protein or polysaccharide.

Although citing the standardized procedure and the vague determination of variation, the study still only reflects results on cells using a single extract.  This speaks to lack of scientific rigor, especially for undefined/uncharacterized extracts. 

Other: (line 33) the extract stimulated a NF-kB-responsive promoter, not the NF-kB promoter itself